# Caffeine-inducible gene switches controlling experimental diabetes

Daniel Bojar [1], Leo Scheller [1], Ghislaine Charpin-El Hamri[2], Mingqi Xie[1] & Martin Fussenegger [1,3]

Programming cellular behavior using trigger-inducible gene switches is integral to synthetic biology. Although significant progress has been achieved in trigger-induced transgene expression, side-effect-free remote control of transgenes continues to challenge cell-based therapies. Here, utilizing a caffeine-binding single-domain antibody we establish a caffeine-inducible protein dimerization system, enabling synthetic transcription factors and cell-surface receptors that enable transgene expression in response to physiologically relevant concentrations of caffeine generated by routine intake of beverages such as tea and coffee. Coffee containing different caffeine concentrations dose-dependently and reversibly controlled transgene expression by designer cells with this caffeine-stimulated advanced regulators (C-STAR) system. Type-2 diabetic mice implanted with microencapsulated, C-STAR-equipped cells for caffeine-sensitive expression of glucagon-like peptide 1 showed substantially improved glucose homeostasis after coffee consumption compared to untreated mice. Biopharmaceutical production control by caffeine, which is non-toxic, inexpensive and only present in specific beverages, is expected to improve patient compliance by integrating therapy with lifestyle.

[1] Department of Biosystems Science and Engineering, ETH Zurich, Mattenstrasse 26, 4058 Basel, Switzerland. [2] IUT, Département Génie Biologique, Institut Universitaire de Technologie, F-69622 Villeurbanne Cedex, France. [3] Faculty of Life Science, University of Basel, Mattenstrasse 26, CH-4058 Basel, Switzerland. Correspondence and requests for materials should be addressed to M.F. (email: fussenegger@bsse.ethz.ch)

In recent years, synthetic biology, the fusion between engineering and biology[1], has brought the rational and predictable construction of sophisticated gene circuits into the forefront of biomedical research. Plug-and-play combinations of carefully designed biological modules have enabled major advances in therapies for personalized medicine[2,3], as well as in the challenging endeavor of lineage control[4,5], bringing achievements in the laboratory ever closer to rewarding real-world applications[6,7]. In this context, cell-based therapies capitalizing on the complexity of mammalian cells are taking the lead in the advent of personalized medicine[8], as exemplified by applications of chimeric antigen receptor (CAR)—T cells[9] or designer cell implants to treat various common diseases[2,10].

Controlling the dynamics of gene expression is essential for the functionality of synthetic gene circuits. This is especially relevant in synthetic biology-inspired therapies, where gene expression regulation determines the dosage of the produced therapeutic and allows for considerable control over the designer cell implant. Therefore, gene expression in most circuits is controlled either at the transcriptional or translational level. At the transcriptional level, promoters responsive to specific triggers are controlled by transcription factors[11], whereas at the translational level, ribozymes or riboswitches responsive to specific triggers control protein translation[12]. In recent years, the quest for better inducers has progressed rapidly. Initially, antibiotics such as tetracycline or doxycycline[13] were used for the control of gene expression, raising issues such as antibiotic resistance[14] and side effects[15]. The next generation of inducers were designed to be safe and orthogonal, such as the apple tree leaf metabolite phloretin[16] or the food additives benzoate and vanillic acid[17]. However, these inducers still suffer from potential side effects, especially in long-term applications, and have to be exogenously added. Traceless inducers, such as light[18] or temperature[19], have recently been developed, but ambient light and ambient temperature make them less orthogonal than would be desirable. The ideal inducer would be inexpensive, would have no side effects, and would be present in only a specific set of known sources.

Here, we report a bioengineering approach for the induction of gene expression in mammalian designer cells by caffeine. The small molecule caffeine is non-toxic[20], cheap to produce[21], and only present in specific beverages, such as coffee and tea. Every day, more than two billion cups of coffee are being consumed worldwide, making coffee one of the most popular beverages after water, and one of the most important cash crops in the world[22]. Currently available caffeine-responsive gene switches require enzymatic conversion of caffeine to theophylline to provide translation control in yeast[23]. However, due to its low sensitivity the yeast system is unsuitable for sensing physiologically relevant caffeine concentrations in mammalian cells. To engineer fully synthetic caffeine-inducible gene switches with user-defined sensitivity and functionality, we established a caffeine-mediated protein dimerization system in mammalian cells based on a single-domain VHH camelid antibody (referred to as aCaffVHH) that has high affinity ($K_d = 500$ nM) and homodimerizes in the presence of caffeine[24–26]. By fusing aCaffVHH to the intracellular signaling domains of different mammalian receptor classes, we created fully synthetic receptors that sense caffeine at physiologically relevant levels (e.g., the amount provided in a cup of coffee). The robustness of these caffeine receptors, which we call C-STAR (caffeine-stimulated advanced regulators), is demonstrated in vitro with pure caffeine and with a diverse array of everyday sources of caffeine, such as black tea, coffee, and energy drinks, as well as in vivo in two mouse models of experimental Type-2 diabetes.

Type-2 diabetes mellitus (T2D) affects more than 400 million people worldwide[27] and associated health costs amount to about 825 billion US dollars per year[28]. As T2D is characterized by a sustained increase in blood glucose levels after each meal, we wanted to capture the natural dynamics of caffeine uptake after each major meal to achieve a novel therapeutic approach to the acute phase of T2D by using designer cells equipped with C-STAR to deliver synthetic human glucagon-like peptide 1 (shGLP-1) in response to dietary intake of coffee or other caffeine-containing beverages. Capitalizing on routine cultural habits, therapies based on such systems should seamlessly integrate into people's lifestyle, and therefore could be a key pillar upon which the new generation of personalized medicine can build.

## Results

**Design of a caffeine-inducible gene switch.** After drinking an average cup of coffee, blood levels of caffeine peak in the low micromolar range[29,30], so for the present purpose, we required a novel caffeine sensor system for non-toxic (Supplementary Fig. 1), physiologically relevant concentrations. To capture these concentrations, we established a caffeine-inducible protein dimerization system in mammalian cells to create different types of gene switches. (i) Fusion of the caffeine-binding single-domain antibody aCaffVHH to DNA-binding and transactivation domains reconstitutes synthetic transcription factors driving chimeric target promoters in a caffeine-responsive manner. (ii) Fusion of the caffeine-binding single-domain antibody aCaffVHH to intracellular signaling domains of different mammalian receptor classes reconstitutes synthetic signaling cascades and allows caffeine to dose-dependently activate different pathway-specific promoters (Fig. 1a).

To design an aCaffVHH-dependent transcription factor-based gene switch, we C-terminally fused aCaffVHH to the DNA-binding TetR-domain ($P_{SV40}$-TetR-aCaffVHH-pA$_{SV40}$, pDB307), as well as N-terminally to four repeats of a transactivating 12-amino-acid peptide (VP$_{min}$, $P_{CAG}$-aCaffVHH-VP$_{min}$x4-pA$_{\beta G}$, pDB335). In this design, the presence of caffeine should dimerize the DNA-binding domain with the transactivating VP$_{min}$ domain and lead to gene expression (Fig. 1b). Utilizing the reporter gene human placental-secreted alkaline phosphatase (SEAP) controlled by a TetR-dependent promoter ($P_{tetO7}$-SEAP-pA$_{SV40}$, pMF111), we observed clear caffeine-dependent gene expression in the presence of 100 μM caffeine (Fig. 1b).

We reasoned that this low sensitivity to caffeine might be due to the absence of signal amplification in this split transcription factor setup. Therefore, we applied the caffeine-inducible dimerization system to different signaling pathway-specific signal transduction domains. First, we fused aCaffVHH N-terminally to the transmembrane domain of interleukin 13 receptor subunit alpha 1 (IL13Rα1, $P_{hCMV}$-aCaffVHH-IL13Rα1-pA$_{bGH}$, pDB323), as well as interleukin 4 receptor subunit alpha (IL4Rα, $P_{hCMV}$-aCaffVHH-IL4Rα-pA$_{bGH}$, pDB324). Addition of caffeine should induce heterodimerization of these receptors and activate signal transducer and activator of transcription 6 (STAT6) signaling. Indeed, when we co-transfected STAT6 ($P_{hCMV}$-STAT6-pA$_{bGH}$, pLS16) and a STAT6-responsive reporter construct ($P_{STAT6}$-SEAP-pA$_{SV40}$, pLS12), we could see caffeine-dependent gene expression starting from 1 μM caffeine (Fig. 1c), a considerable improvement in sensitivity compared to the split transcription factor setup using pDB307 and pDB335. However, the absolute output strength of this setup in SEAP units was limited, necessitating a more powerful system.

To overcome the output strength issue, we fused aCaffVHH C-terminally to the intracellular part of the murine fibroblast growth factor receptor 1 (mFGFR1, $P_{hCMV}$-mFGFR1$_{405-822}$-aCaffVHH-pA$_{bGH}$, pDB395)[31]. The presence of caffeine should

homodimerize mFGFR1$_{405-822}$-aCaffVHH and lead to MAPK signaling, which we re-routed to TetR-dependent pMF111 by co-transfecting TetR-Elk1 (P$_{hCMV}$-TetR-Elk1-pA$_{bGH}$, MKp37). The signal amplification of the MAPK signaling cascade[32] yielded a strong and sensitive gene expression response in the presence of as little as 0.01 µM caffeine (Fig. 1d). However, this extraordinary sensitivity to caffeine may be detrimental in a therapeutic setting, since even trace amounts of caffeine would induce the gene circuit. Additionally, the requirement of the re-routing protein TetR-Elk1 meant that transfection of three plasmids was necessary for this system.

Improving on the mFGFR1-dependent system, we fused aCaffVHH N-terminally to an erythropoietin receptor derivative (EpoR, P$_{hCMV}$-aCaffVHH-EpoR$_m$-IL-6RB$_m$-pA$_{bGH}$, pDB306)[33,34], leading to homodimerization of the receptor in the presence of caffeine and subsequent JAK/STAT signaling through STAT3. As HEK-293T cells endogenously express STAT3, we only needed to transfect pDB306 and a STAT3-dependent reporter plasmid (P$_{STAT3}$-SEAP-pA$_{SV40}$, pLS13). This setup yielded a strong and sensitive gene expression system with a maximal response at 1 µM caffeine (Fig. 1e).

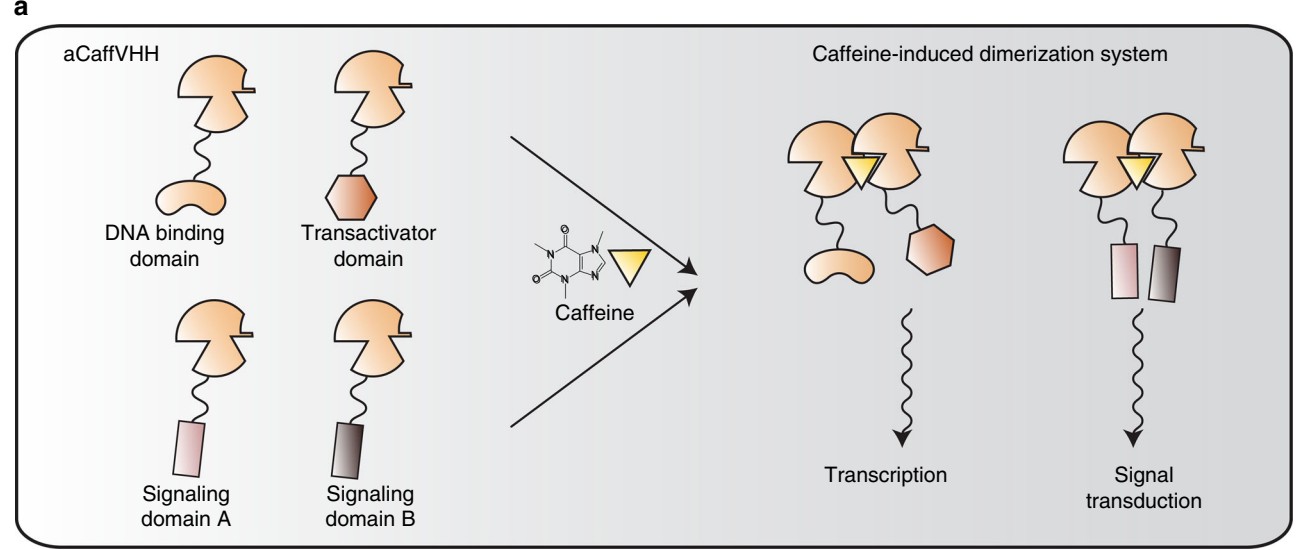

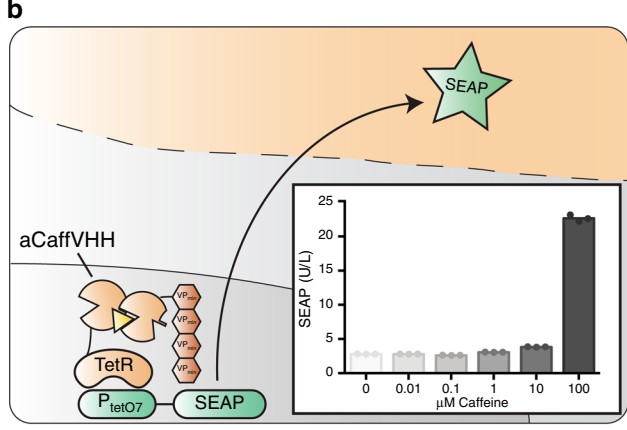

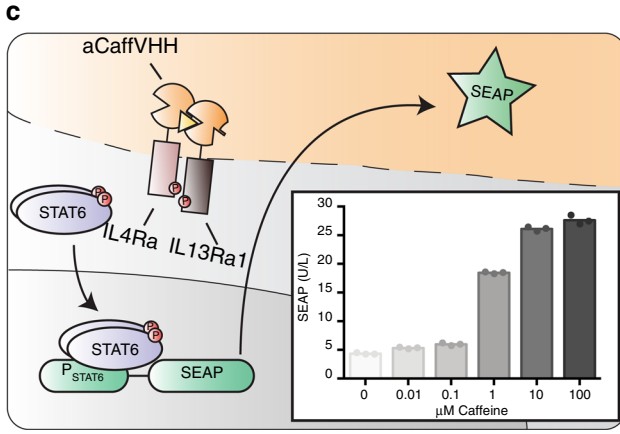

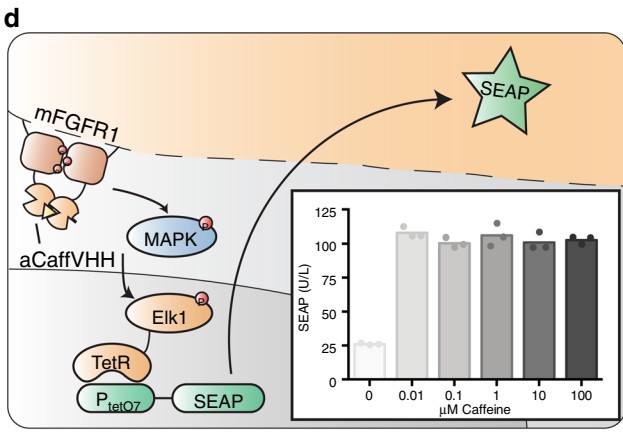

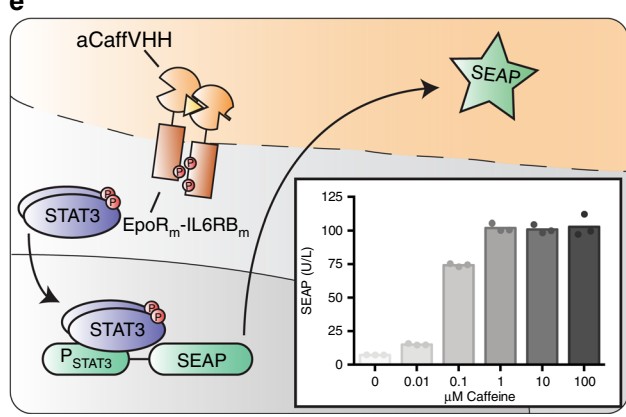

Overall, caffeine-dependent STAT3-signaling proved to be the best fit in terms of potency, sensitivity to physiological caffeine levels, and number of components, and so it was used for all further experiments. Due to receptor homodimerization and endogenous STAT3 expression, we only needed to transfect two components to obtain a full C-STAR system. Since the presented gene expression systems had different sensitivities and relied on orthogonal promoters, they could be used for endowing designer cells with a nonlinear response to caffeine by expressing multiple receptors (Supplementary Fig. 2a, b).

### Characterization of the caffeine-inducible C-STAR system.

Functionality of the C-STAR system was also demonstrated in human telomerase reverse transcriptase-immortalized human mesenchymal stem cells (hMSC-hTERT) (Fig. 2a). However, HEK-293T cells showing higher caffeine sensitivity and protein secretion capacity were used in all follow-up experiments. For long-term experiments, the C-STAR receptor ($P_{hEF-1\alpha}$-aCaffVHH-EpoR$_m$-IL-6RB$_m$-pA$_{SV40}$, pDB326) was stably integrated into the genome of HEK-293T cells, creating the designer cell line C-STAR$_{DB1}$. The caffeine dose-response relationship of this polyclonal cell line was similar to that of the transiently transfected cells (Fig. 2b). However, selection of monoclonal C-STAR cell lines yielded clones with different sensitivities for caffeine (Supplementary Fig. 3a–d). All further in vitro experiments were conducted with the C-STAR$_{DB1}$ cell line.

To capture the time window of high caffeine concentration in the blood, an in vivo C-STAR system would need to induce gene expression after brief exposure to the inducer. Exposure to physiologically relevant concentrations of caffeine induced a half-maximal response of the C-STAR system within just one hour, and a maximal response was obtained after six hours of exposure (Fig. 2c). Since the half-life of caffeine in human blood is approximately five hours[30], in vivo activation of the C-STAR system by caffeine should be feasible. Among the caffeine analogs tested in vitro, only theophylline showed modest cross-activation of C-STAR at 1 µM concentration (Supplementary Fig. 4), which is unlikely to be reached in the physiological situation[35,36]. The response time of the C-STAR system after caffeine addition was assessed and the C-STAR system responded in a timely manner to the presence of caffeine, yielding detectable amounts of reporter protein at 12 h, whereas no induction of SEAP expression was seen in the negative control lacking caffeine (Fig. 2d). Testing the reversibility of the gene circuit, C-STAR$_{DB1}$ cells were incubated with physiologically relevant concentrations of caffeine or the equivalent amount of H$_2$O (mock), with an exchange of caffeine to mock, or vice versa, every day (Fig. 2e). As expected, the system was shut off by the removal of caffeine and

could be activated again by the renewed addition of caffeine to the cells, indicating reversibility after removal or degradation of caffeine.

### Caffeine quantification in commercial beverages using C-STAR.

Caffeine is a component of various beverages. Therefore, to broaden the range of available beverages for the induction of the C-STAR system, and to establish the specificity of the synthetic biology-inspired caffeine-sensing system, C-STAR$_{DB1}$ cells were challenged with 26 products, including Nespresso Grand Cru®, Starbucks® coffee, Red Bull®, Cuida Te® tea capsule, and Coca-Cola® (Fig. 3a). Several Nespresso Grand Cru® capsules were also tested in their decaffeinated version as negative controls (Vivalto lungo decaffeinato®, Volluto decaffeinato®, Decaffeinato intenso®, and Arpeggio decaffeinato®). As three of these beverage samples also have caffeinated versions (Vivalto lungo®, Volluto®, and Arpeggio®), which are claimed by the manufacturer to be identical to the respective decaffeinated versions except for the caffeine content, they allowed us to confirm that caffeine itself upregulates gene expression and not any other of the hundreds of chemical compounds present in coffee[22]. Overall, our beverage samples covered a wide range of caffeine concentrations from 0 to 4.8 g L$^{-1}$. A standard dose-response curve was obtained with pure caffeine. This enabled us to convert the SEAP values from C-STAR$_{DB1}$ cells incubated with beverage samples into caffeine concentrations.

For all samples tested, caffeine concentrations indicated by the vendor corresponded well to those measured with C-STAR$_{DB1}$ cells (Fig. 3b, c). As expected, decaffeinated beverage samples showed very low activation of the C-STAR system (Fig. 3b, c). These results indicate that C-STAR reproducibly generates a dose-dependent, caffeine-specific response.

### C-STAR treatment for obesity-induced Type-2 diabetes.

The functionality of the designed C-STAR system in vascularized microcontainers was first confirmed in vitro with pure caffeine (Supplementary Fig. 5). After validating the immunoprotective function of microcapsule implants for drug delivery in vivo (Supplementary Fig. 6), mice implanted with the designer cell capsules were given room temperature Volluto® coffee (Nespresso Grand Cru®), or H$_2$O to drink. As expected, only mice grafted with the C-STAR system showed reversible, coffee-induced SEAP expression (Supplementary Fig. 7a, b). The same mice were re-stimulated a few days later and showed the same response as in the initial experiment (Supplementary Fig. 7c, d).

Next, in order to examine whether this system could be utilized for caffeine-induced treatment of obesity-induced T2D, we replaced the reporter gene SEAP with the gene coding for

**Fig. 1** Synthetic biology-inspired genetic circuits for caffeine-induced gene expression. **a** Caffeine-inducible protein dimerization system based on the camelid-derived single-domain antibody aCaffVHH. aCaffVHH homodimerizes in the presence of caffeine and can be used to reconstitute synthetic transcription factors or signaling cascades that fine-tune caffeine-responsive gene expression. **b** Caffeine-sensing circuit based on the heterodimerization of aCaffVHH-TetR (pDB307) and aCaffVHH-VP$_{minx4}$ (pDB335), leading to direct transcriptional activation. The caffeine dose-response relationship was quantified with the reporter gene SEAP ($P_{tetO7}$-SEAP-pA$_{SV40}$, pMF111). **c** Caffeine-sensing circuit based on the IL13 receptor and the JAK/STAT6 pathway. Caffeine-induced heterodimerization of aCaffVHH-IL13Rα1 (pDB323) and aCaffVHH-IL4Rα (pDB324) leads to phosphorylation of STAT6 (pLS16) by JAK kinases and subsequent transcriptional activation of the STAT6-responsive promoter $P_{STAT6}$. The caffeine dose-response relationship was quantified with the reporter gene SEAP ($P_{STAT6}$-SEAP-pA$_{SV40}$, pLS12). **d** Caffeine-sensing circuit based on the MAPK pathway. Caffeine-induced homodimerization of mFGFR1$_{405-822}$-aCaffVHH (pDB395) led to phosphorylation of MEK1/2 and downstream signaling of the MAPK cascade. Rewiring the signaling cascade through the hybrid transcription factor TetR-Elk1 (MKp37) led to expression of the reporter gene SEAP ($P_{tetO7}$-SEAP-pA$_{SV40}$, pMF111), enabling quantification of the caffeine dose-response relationship. **e** Caffeine-sensing circuit based on the Epo receptor and the JAK/STAT3 pathway. Caffeine-induced homodimerization of aCaffVHH-EpoR$_m$-IL-6RB$_m$ (pDB306) leads to phosphorylation of STAT3 by JAK kinases and subsequent transcriptional activation of the STAT3-responsive promoter $P_{STAT3}$. The caffeine dose-response relationship was quantified with the reporter gene SEAP ($P_{STAT3}$-SEAP-pA$_{SV40}$, pLS13). Data in (**b–e**) are shown as the mean in bar graphs and symbols indicate individual data points. The data displayed represent three independent experiments ($n = 3$)

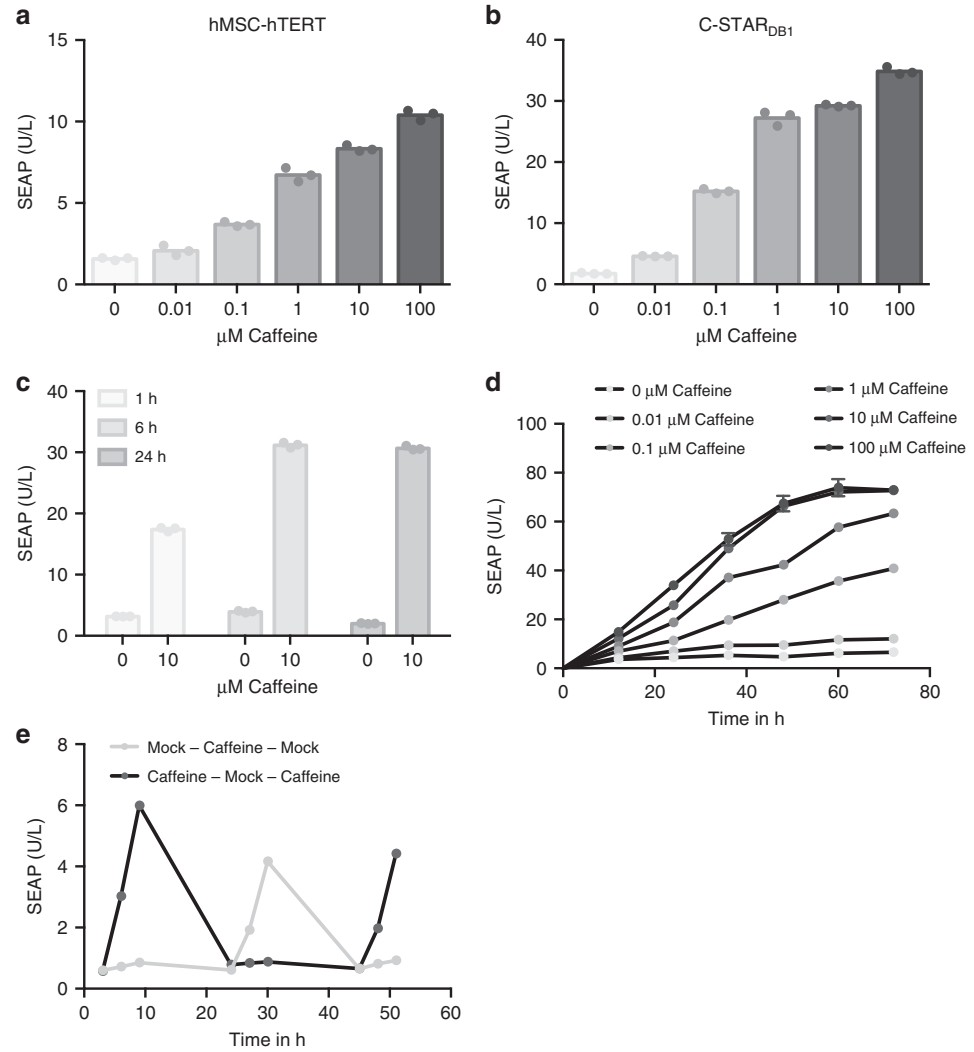

**Fig. 2** Characterization of the caffeine-induced gene switch C-STAR. **a** Functionality of C-STAR in hMSC-hTERT cells. hMSC-hTERT cells were transiently transfected with pDB306 ($P_{hCMV}$-aCaffVHH-EpoR$_m$-IL-6RB$_m$-pA$_{bGH}$) and pLS13 ($P_{STAT3}$-SEAP-pA$_{SV40}$). Sixteen hours after transfection, the cells were exposed to increasing concentrations of caffeine in standard cell culture medium. The caffeine dose-response relationship was quantified in terms of SEAP expression after 24 h. The data displayed represent three independent experiments ($n = 3$). **b** Caffeine-responsiveness of polyclonal C-STAR$_{DB1}$ cells. Polyclonal C-STAR$_{DB1}$ cells were exposed to increasing caffeine concentrations to examine their sensitivity. Supernatant levels of SEAP were quantified after 24 h. The data displayed represent three independent experiments ($n = 3$). **c** Caffeine exposure time needed for the activation of the C-STAR system. C-STAR$_{DB1}$ cells were exposed to H$_2$O or 10 µM caffeine in standard cell culture medium for different periods of time to determine the minimum exposure time needed for induction. After the indicated time, the caffeinated medium was replaced with standard cell culture medium and SEAP expression proceeded for 24 h before quantification. Data in **a**–**c** are shown as the mean in bar graphs and symbols indicate individual data points. The data displayed represent three independent experiments ($n = 3$). **d** Response time of the C-STAR system to caffeine. C-STAR$_{DB1}$ cells were exposed to H$_2$O or increasing concentrations of caffeine in standard cell culture medium to determine the response time of the system. Supernatant samples containing SEAP were taken every 12 h for 72 h. The data displayed represent the means ± s.d. of three independent experiments ($n = 3$). **e** Reversibility of the C-STAR system. C-STAR$_{DB1}$ cells were alternately exposed to H$_2$O and 10 µM caffeine in standard cell culture medium to show the reversibility of the system. Supernatant samples containing SEAP were taken every three hours for nine hours per day. The data displayed represent the means ± s.d. of three independent experiments ($n = 3$)

synthetic human glucagon-like peptide coupled to mouse IgG (shGLP-1, $P_{STAT3}$-shGLP-1-pA$_{SV40}$, pDB387), an engineered protein clinically licensed for the treatment of T2D[37]. Experiments in vitro with the C-STAR$_{DB6}$ cell line incorporating the resulting construct validated the caffeine-dependent expression of shGLP-1 (Supplementary Fig. 8a, b). Pharmacokinetic analyses of caffeine and shGLP-1 in mice confirmed the potential of C-STAR$_{DB6}$ for cell-based diabetes therapy; a single oral administration of coffee resulted in a transient surge of caffeine in the bloodstream[38] that was sufficient to trigger sustained shGLP-1 activity (Supplementary Fig. 9). Importantly,

hypoglycemic side effects were not observed following higher levels of caffeine-dependent shGLP-1 production (Supplementary Fig. 10), confirming the inherent inactivity of GLP-1 in normoglycemic environments[39,40]. Then, we examined the efficacy of these cells in two T2D mouse models with impaired insulin sensitivity. For this purpose, diet-induced obesity[41] (DIO; Fig. 4) and leptin receptor-deficient[41] (db/db; Fig. 5) mice were implanted intraperitoneally with capsules containing C-STAR$_{DB6}$ cells or with control capsules containing cells equipped with only the output module pDB387 (mock). All mice received regular oral doses of Volluto® coffee. DIO mice treated with C-STAR$_{DB6}$ cells

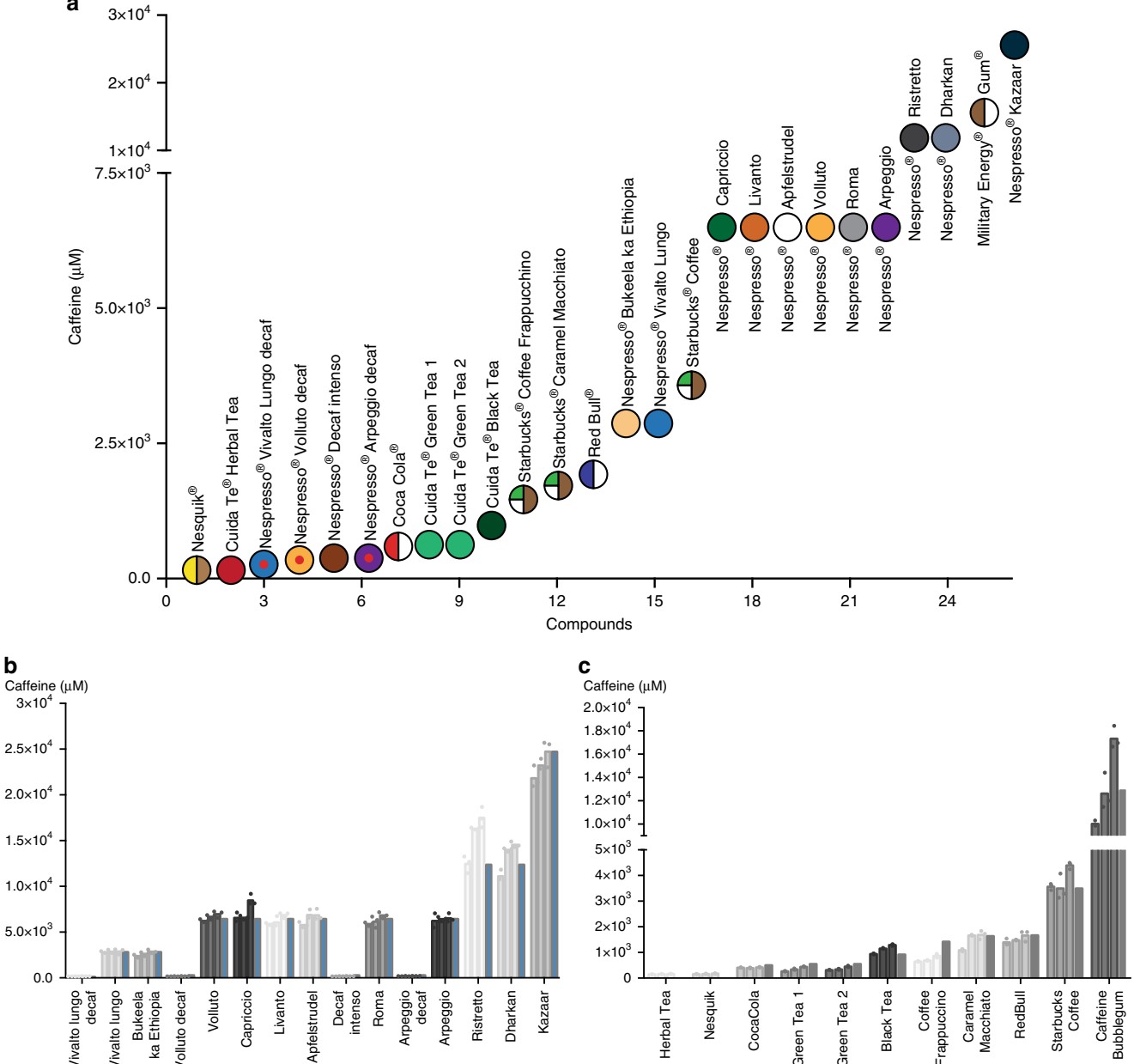

**Fig. 3** In vitro caffeine quantification in commercial caffeine sources. **a** Illustration of the tested solutions with their respective caffeine concentration. From left to right, the boxes correspond to Nesquik® capsules, Forest Fruits® (herbal tea), Vivalto lungo decaffeinato®, Volluto decaffeinato®, Decaffeinato intenso®, Arpeggio decaffeinato®, Coca-Cola®, Mediterranean® (green tea), Marrakech® (green tea), Earl Grey® (black tea), Starbucks® Coffee Frappuccino, Starbucks® Caramel Macchiato, Red Bull®, Bukeela ka Ethiopia®, Vivalto lungo®, Starbucks® Coffee, Capriccio®, Livanto®, Apfelstrudel®, Volluto®, Roma®, Arpeggio®, Ristretto®, Dharkan®, Military Energy Gum®, and Kazaar®. The indicated caffeine concentrations were calculated from the specifications of the vendor regarding the amount of caffeine in each beverage. **b, c** Quantification of the caffeine concentration in coffee from Nespresso Grand Cru® capsules (**b**) and other commercially available caffeine sources (**c**). Caffeine-containing samples were added to C-STAR_{DB1} cells with a dilution of 1:50,000. A standard curve obtained with pure caffeine enabled conversion of the quantified SEAP levels to caffeine concentrations in the original samples. Each beverage was prepared or bought on three separate occasions and the data represent the quantification of each replicate in triplicate ($n = 3$). Data in (**b, c**) are shown as the mean in bar graphs and symbols indicate individual data points. The caffeine concentration indicated by the vendor is shown in blue

exhibited lower fasting blood glucose values throughout a two-week experimental time course compared to the untreated control group (Fig. 4a). To demonstrate improved glycemic control in C-STAR_{DB6}-treated T2D mice, a glucose tolerance test was conducted to simulate a meal response. As expected, C-STAR_{DB6}-triggered GLP-1 production (Fig. 4b) increased the insulin levels of DIO mice (Fig. 4c) and established near-

homeostatic postprandial glucose metabolism in coffee-treated diabetic mice (Fig. 4d). For db/db mice, which develop increased hyperinsulinemia compared to DIO mice[42] (Figs. 4c and 5b), GLP-1-dependent insulinotropic action (Fig. 5a, b) and glucose tolerance (Fig. 5c) were also restored, but required a higher dose of implanted C-STAR_{DB6} cells (Fig. 5a–c). Importantly, this coffee-triggered C-STAR_{DB6}-based diabetes therapy did not

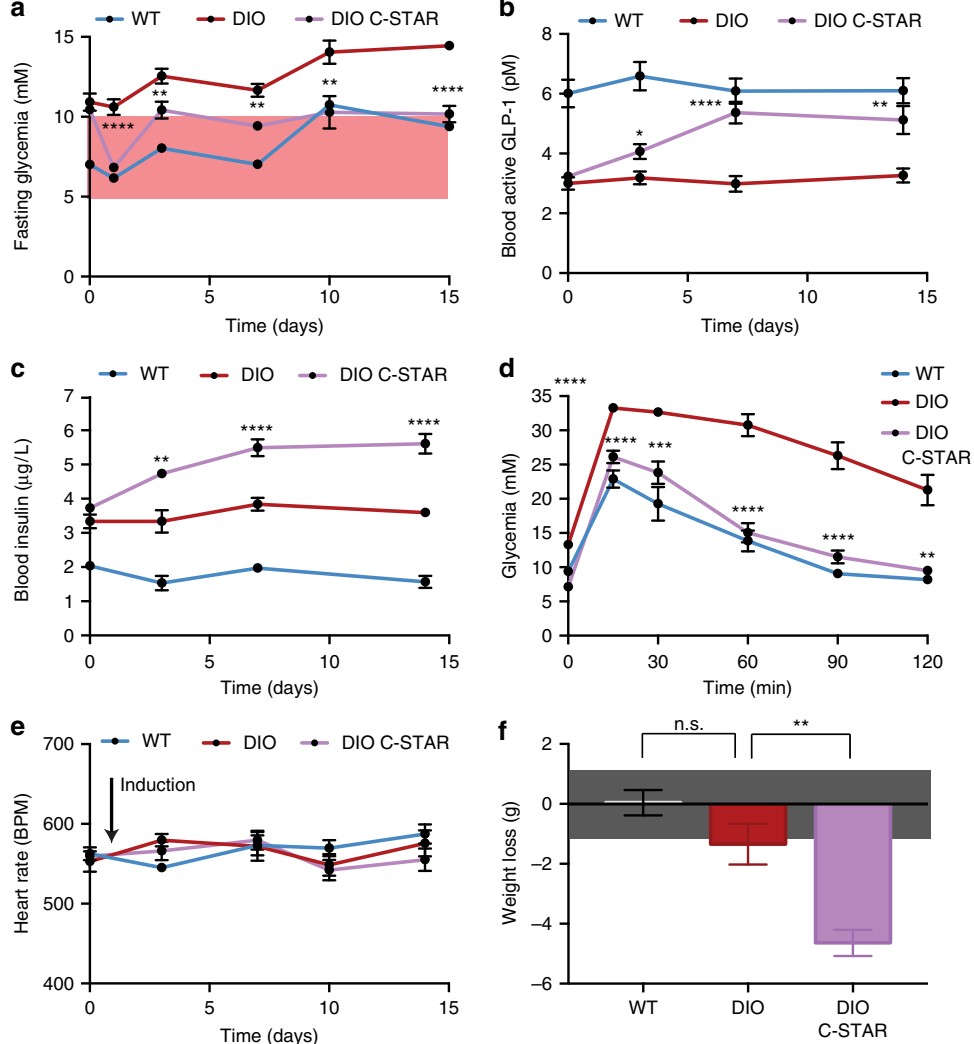

**Fig. 4** Coffee-induced designer cell-based treatment of diabetic diet-induced obese mice. **a–d** Caffeine-dependent insulinotropic action of shGLP-1. Wild type (WT) or diet-induced obese mice (DIO) were intraperitoneally implanted with microencapsulated C-STAR$_{DB6}$ cells or control HEK-293T cells containing only pDB387 (P$_{STAT3}$-shGLP-1-pA$_{SV40}$) and received daily oral doses of 300 μL Nespresso Volluto® coffee. **a** Fasting glycemia, **b** blood active GLP-1, and **c** 4 h postprandial insulin levels were recorded for 14 days. **d** Intraperitoneal glucose tolerance tests were performed by administration of 2 g kg$^{-1}$ aqueous D-glucose. **e** Caffeine-dependent cardiovascular effects. Heart rate of the same mice shown in (**b**, **c**) was measured prior to the collection of blood samples. **f** Caffeine-triggered shGLP-1-mediated effects on body weight. On day 15, the body weights of individual mice shown in (**a–e**) were compared to their initial body weights (day 1; prior to first coffee intake). The confidence interval of the balance is indicated by a gray box. All data displayed are mean ± SEM ($n = 10$ mice). Comparisons were made with Welch's $t$ test: *$P < 0.05$, **$P < 0.01$, ***$P < 0.001$, ****$P < 0.0001$ vs. control, n.s. not significant. The range of homeostatic fasting glycemia is indicated with a red box

impact on the heart rate of treated animals (Figs 4e and 5d), but reduced the body weight of diet-induced obese mice after 2 weeks (Fig. 4f).

## Discussion

The C-STAR system developed here extends previous efforts[23] to induce gene expression with caffeine by enabling engineered mammalian cells to directly sense caffeine at physiologically relevant concentrations, thereby making it possible to fine-tune therapeutic transgene expression in response to routine intake of beverages, such as tea and coffee without supplementation of any additional chemicals. Receptor setups with differing sensitivity (Fig. 1), as well as different monoclonal cell lines generated from the C-STAR$_{DB1}$ system (Supplementary Fig. 3), could be useful to accommodate different lifestyles of patients, who may consume different amounts of caffeine per day. As the C-STAR system

responds dose-dependently to caffeine, a variety of caffeine-containing beverages, ranging from coffee or tea to energy drinks, can be used to trigger the system. Importantly, decaffeinated coffee did not activate the C-STAR system, so C-STAR-treated patients could still enjoy decaffeinated drinks without activating their implants. Additionally, coffee capsules such as Nespresso Grand Cru® are highly standardized and could allow for predictable dosing. The persistence, efficacy, and immunoprotective functions of alginate-based microcontainers used in this study have already been demonstrated in diabetes clinical trials, paving the way for the application of C-STAR cells in humans[43].

Potential side effects of caffeine are minimal and well known, even in the case of lifelong consumption[44,45]. Indeed, normal doses of caffeine in the form of coffee are reported to have health benefits[46–49]. Even caffeine doses of up to 400 mg per day have no adverse effect in adults[20], so that a broad range of caffeine consumption is available for the control of C-STAR cells. As a natural

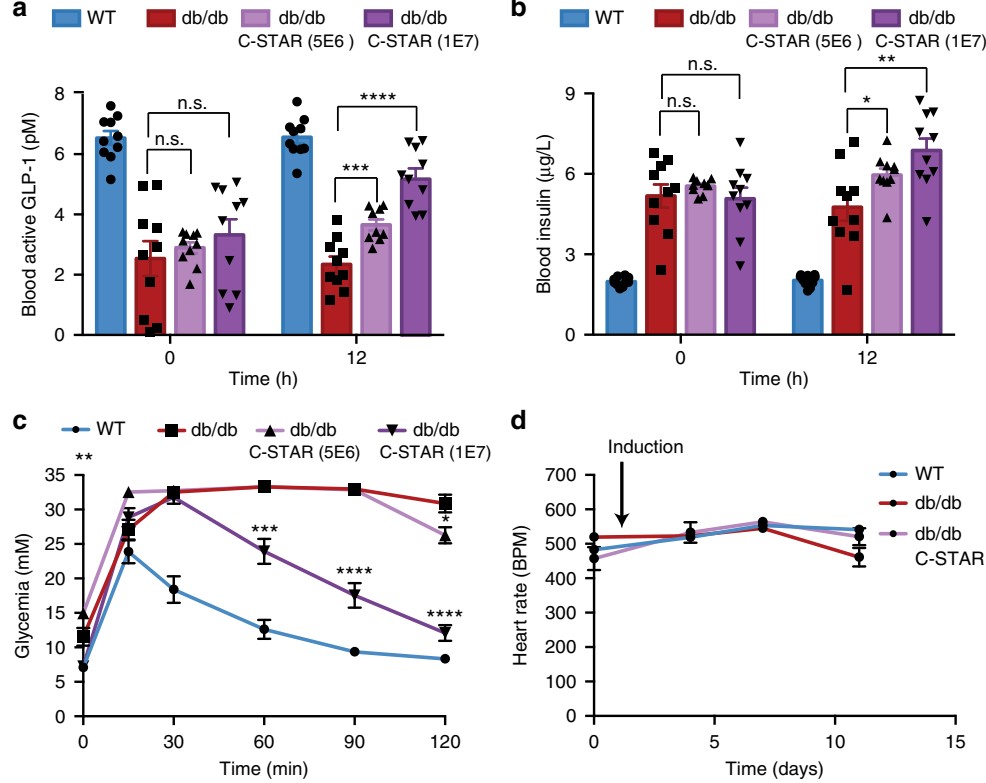

**Fig. 5** Coffee-induced designer cell-based treatment of diabetic db/db mice. **a–c** Caffeine-dependent insulinotropic action of shGLP-1. Wild type (WT) or leptin receptor-deficient mice (db/db) were intraperitoneally implanted with different doses of microencapsulated C-STAR$_{DB6}$ cells (0 to $1 \times 10^7$ cells) or $1 \times 10^7$ control HEK-293T cells containing only pDB387 ($P_{STAT3}$-shGLP-1-pA$_{SV40}$), and received an oral dose of 300 μL Nespresso Volluto® coffee. **a** blood active GLP-1 and **b** 4 h postprandial insulin levels were recorded before cell implantation and 1 day afterwards. **c** Intraperitoneal glucose tolerance tests were performed by administration of 2 g kg$^{-1}$ aqueous D-glucose. **d** Caffeine-dependent cardiovascular effects. Heart rate of the same mice described in (**a–c**) was measured prior to the collection of blood samples. All data displayed are mean ± SEM ($n = 10$ mice). Comparisons were made with Welch's $t$ test: *$P < 0.05$, **$P < 0.01$, ***$P < 0.001$, ****$P < 0.0001$ vs. control, n.s. not significant

ingredient of beverages, caffeine (unlike most other chemical inducers) is a popular stimulant consumed by a large proportion of the population[50]. Caffeine is cheap and easily synthesized[21], making it far more cost-effective than, for instance, non-immunosuppressive analogs of the popular inducer rapamycin[51]. Major causes of patient noncompliance (i.e., failure of patients to follow medication instructions)[52] are complicated instructions, forgetfulness, and disruption of lifestyle[53]. As even prevalent diseases, such as Type-2 diabetes, are associated with high levels of noncompliance[54], an inducer that is present in routinely consumed beverages could be highly beneficial. On all these grounds, we think caffeine is a promising candidate in the quest for the most suitable inducer of gene expression. Additionally, we think that the caffeine-dimerizable single-domain antibody aCaffVHH will be a valuable addition to the range of small-molecule-mediated-dimerization kits due to its small size, high affinity for caffeine, and the feasibility of using it intracellularly, as well as extracellularly in any fusion orientation.

Personalized medicine, the custom-tailored interplay between diagnostics and therapy, has long been predicted[55], but still remains on the horizon. To achieve truly personalized treatment, designed systems need to be highly tunable, so that they can easily be adapted to each individual patient and his or her lifestyle. Any lifestyle disruption would not only impair the quality of life of the patient, but also increase the chances of noncompliance. We believe the tunability and sensitivity of the synthetic biology-inspired C-STAR systems developed here meet these requirements, and these systems are promising candidates for the control of T2D. It is also worth noting that the C-STAR system could even be used as an in vitro caffeine-quantifying device for patients suffering from caffeine hypersensitivity[56].

## Methods

**Plasmid construction.** See table in Supplementary Data 1.

**Cell culture.** Human embryonic kidney cells (ATCC: CRL3216, HEK-293T) and adipose tissue-derived human telomerase reverse transcriptase-immortalized human mesenchymal stem cells (ATCC: SCRC4000, hMSC-hTERT) were cultured in Dulbecco's modified Eagle's medium (DMEM; Life Technologies, Carlsbad, CA, USA) supplemented with 10% (v/v) fetal calf serum (FCS; BioConcept, Allschwil, Switzerland; lot no. 022M3395) and 1% (v/v) penicillin/streptomycin solution (Sigma-Aldrich, Munich, Germany). All cells were cultured in a humidified atmosphere containing 5% $CO_2$ at 37 °C. Cell viability and number was assessed with an electric field multi-channel cell-counting device (CASY Cell Counter and Analyzer Model TT; Roche Diagnostics GmbH, Basel, Switzerland). For transfection in a 24-well plate format, 500 ng of plasmid DNA were diluted in 50 μL FCS-free DMEM, mixed with 2.5 μL polyethyleneimine (PEI; Polysciences Inc.; 1 mg mL$^{-1}$), and incubated at room temperature for 20 min. Then, the transfection mixture was added dropwise to $1.25 \times 10^5$ cells seeded 12 h before transfection. Twelve hours after transfection, the transfection medium was replaced by standard culture medium or medium supplemented with caffeine (cat. No. C0750, Sigma-Aldrich) or caffeine-containing compounds. Transgene expression was profiled 24 h later.

**Generation of genetically stable designer cell lines.** To develop stable designer cell lines according to the Sleeping Beauty transposon protocol[57], one well of a 6-well plate with HEK-293T cells was co-transfected with pDB326 (1900 ng)/pSB100x (100 ng). After 12 h, the transfection medium was exchanged for standard culture medium. After an additional 24 h, the medium was exchanged for standard culture medium supplemented with 1 μg mL$^{-1}$ puromycin (cat. no. A1113803; ThermoFisher Scientific, Reinach, Switzerland) and a polyclonal cell population (C-STAR$_{DB1}$) was selected for 2 weeks. Subsequently, single cells were sorted by FACS according to fluorescence intensity, and single clones were grown in

conditioned HEK-293T culture medium. Monoclonal cell populations were screened for caffeine-responsive SEAP expression and $CSTAR_{DB3}$ was chosen as the best performer. The polyclonal stable cell line $C-STAR_{DB6}$ was similarly generated with the plasmid pDB387, using $100\,\mu g\,mL^{-1}$ zeocin (cat. no. R25005; ThermoFisher Scientific, Reinach, Switzerland) as the selecting reagent.

**CCK-8 assay.** Cell viability was quantified with the Cell Counting Kit-8 (CCK-8; Dojindo Laboratories; cat. no. CK04) according to the manufacturer's instructions in Corning® 96 black well plates with a clear bottom (cat. no. CLS3603, Sigma-Aldrich). Briefly, 12 h after transfection, standard culture medium supplemented with or without caffeine was added to the cells. After 24 h, the medium was exchanged for standard culture medium supplemented with 10% (v/v) Cell Counting Kit-8. After an incubation period of one hour at 37 °C, absorbance was measured at 450 nm with an EnVision 2104 multilabel reader (PerkinElmer), yielding a surrogate for cell viability.

**SEAP assay.** For the quantification of human placental-secreted alkaline phosphatase (SEAP), cell culture supernatant was heat-inactivated for 30 min at 65 °C. Then, 80 μL supernatant was mixed with 100 μL 2 × SEAP buffer (20 mM homo-arginine, 1 mM $MgCl_2$, 21% (v/v) diethanolamine, pH 9.8) and 20 μL of substrate solution containing 20 mM pNPP (Acros Organics BVBA). Measurement was then performed at 405 nm using an EnVision 2104 multilabel reader (PerkinElmer). SEAP production in vivo was quantified with the chemiluminescence SEAP reporter gene assay (cat. no. 11779842001, Sigma-Aldrich) according to the manufacturer's instructions.

**Mouse IgG ELISA.** Mouse IgG levels in samples containing shGLP1-mIgG were quantified using the Mouse IgG ELISA Kit (cat. no. E-90G, ICL Lab), according to the manufacturer's instructions. The absorbance was quantified at 450 nm with an EnVision 2104 multilabel reader (PerkinElmer) and the mouse IgG levels were interpolated with a standard curve.

**Glucose tolerance test.** Mice were challenged by intraperitoneal injection of glucose $(2\,g\,kg^{-1}$ body weight in $H_2O)$ and the glycemic profiles were generated by measurement of blood glucose levels with a glucometer (Contour® Next; Bayer HealthCare, Leverkusen, Germany) every 15 or 30 min for 120 min.

**Insulin ELISA.** Insulin blood levels in tested mice were assessed with the Ultra-sensitive Mouse Insulin ELISA (cat. no. 10-1132-01, Mercodia) according to the manufacturer's instructions. The absorbance was quantified at 450 nm with an EnVision 2104 multilabel reader (PerkinElmer).

**shGLP-1 ELISA.** Blood levels of GLP-1 in tested mice were measured with the High Sensitivity GLP-1 Active ELISA Kit, Chemiluminescent (cat. no. EZGLPHS-35K, Merck) according to the manufacturer's instructions. The absorbance was quantified at 450 nm with an EnVision 2104 multilabel reader (PerkinElmer).

**Caffeine samples.** Coffee (Nespresso Grand Cru®) and tea samples (Cuida Te®) were prepared on a Nespresso Capri Automatic Sand machine (Koenig®). Star-bucks coffee samples were obtained from a local Starbucks®. Coca-Cola® and Red Bull® samples were purchased from a local supermarket. Nesquik® (Nescafé Dolce Gusto®) was prepared on a Circolo Automatic EDG605B EX:1 (Nescafé Dolce Gusto®, DeLonghi). Military energy gum® (MarketRight Inc.) was mechanically crushed, covered with 40 mL water, and shaken for several hours at 37 °C to simulate chewing. Unless indicated otherwise, volumes of prepared beverages were those recommended by the respective manufacturer. All samples were diluted 1:50,000 in standard culture medium and added to the designer cells for quantification of caffeine.

**Animal experiments.** Encapsulated HEK-293T and C-STAR-derivative cells for the intraperitoneal implants were generated with an Inotech Encapsulator Research Unit IE-50R (EncapBioSystems Inc., Greifensee, Switzerland). Coherent alginate-poly-(L-lysine)-beads (400 μm diameter, 500 cells per capsule) were generated with the following parameters: 200-μm nozzle with a vibration frequency of 1025 Hz; 25-mL syringe operated at a flow rate of 410 units; 1.12 kV bead dispersion voltage[58]. Female C57BL/6 (14 weeks old) or T2D mice were injected with 1–2 mL of serum-free DMEM containing $1 \times 10^4$ capsules. As genetically disposed T2D mice, db/db mice (female, 8 weeks old) were purchased from Janvier Labs. For the diet-induced obesity (DIO) model of T2D, C57BL/6 J mice (Janvier Labs, female, 4 weeks old) were fed for 10 weeks with a 10-kcal% or a 60-kcal% fat diet (TestDiet, cat. no. T-58Y1-58126) before C-STAR-controlled treatment. Blood glucose concentration was measured with a glucometer (Contour® Next; Bayer HealthCare, Leverkusen, Germany). Serum was collected using microtainer serum separating tubes (cat. no. 365967; Becton Dickinson, Plymouth, UK) according to the manufacturer's instructions. Experiments involving animals were carried out in accordance with the directive of the European Union by Ghislaine Charpin-El

Hamri (No. 69266309; project No. DR2013–01 (v2)) at the Institut Universitaire de Technologie, UCB Lyon 1, F-69622 Villeurbanne Cedex, France.

**Data availability**. Requests for materials should be made to the corresponding author. All plasmids generated in this study are available upon request. All data are available upon reasonable request.

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

## Acknowledgements

We thank Pratik Saxena, David Fuchs, and Ryosuke Kojima for their generous advice and critical comments on the manuscript. We thank Ted Abel for providing MKp37. This work was supported by the National Centre of Competence in Research (NCCR), Molecular Systems Engineering.

## Author contributions

D.B., M.X., and M.F. designed the project, analyzed the results, and wrote the manuscript, D.B. and L.S. conducted the in vitro work, G.C.-E.H. performed the animal experiments.

## Additional information

**Competing interests:** The authors declare no competing interests.

