## [Peer Review File · Nature Communications]

Reviewers' comments:

Reviewer #1 (Remarks to the Author):

In this manuscript, Bojar and colleagues designed and evaluated a series of synthetic factors that upon transfection into the host cell induce a transient activation of targeting signaling events upon induction by caffeine. Using a series of elegant in vitro and in vivo approaches, the authors convincingly show caffeine-induction of targeted signaling events, suggesting that such a caffeine-induced approach has a broad pharmaceutical value for the treatment of various metabolic conditions.

The manuscript is well written, easy to follow and deals with a novel, innovative and important topic, the design and validation of specially-tailored molecules/constructs aimed to optimize the therapeutic value of certain drugs. The manuscript is considered as of importance for its field.

Main criticism:

- the authors state to have analyzed the caffeine-inducible GLP-1 construct in a mouse model for type 2 diabetes. The authors describe to have generated this T2DM model by repeated administration of streptocotozin (STZ). Notably, STZ is not inducing type 2 diabetes, it rather drives apoptosis of the beta cells and thus the development of hypoinsulinemia and type 1 diabetes. Type 2 diabetes is typically characterized by hyperinsulinemia and insulin resistance but the authors here see, as expected by STZ treatment, hypoinsulinemia and no insulin resistance. So this model can't be referred to as type 2 diabetic. Typically db/db mice or NZO mice are considered the best available mouse models for type 2 diabetes but not STZ treated mice.

- Apart from the misclassification of the STZ mice as type 2 diabetic, these mice develop a somehow unexpected phenotype that warrants clarification. In detail, according to Figure 4f, fasting levels of blood glucose are roughly 145mg/dl (around 8 mM/L), which is still in the physiological range of non-STZ treated mice and is way lower as what is expected from a mouse that has repeatedly received an STZ treatment and which is hypoinsulinemic (as demonstrated in Figure 4d). Also, since GLP-1 decreases blood glucose via stimulation of insulin secretion, the insulinotropic action of GLP-1 is typically blunted in hypoinsulinemic type 1 diabetic STZ treated mice. To consolidate their findings, it is suggested that the authors repeat the in vivo study in an established mouse model for type 2 diabetes, like the db/db or the NZO mouse. Analysis of the construct in diet-induced obese (DIO) mice would further be of appreciable merit. Does caffeine-induced GLP-1 secretion decrease body weight in obese DIO mice? Especially the in vivo part can be substantially improved and would help to underline the pharmacological value of the system. Given the cardiovascular effect of caffeine, a control group of WT animals getting caffeine (but in which the caffeine can't activate the GLP-1 construct) would be important to add in these studies.

Minor criticism

- since the construct needs to be implanted into the host organism, I would appreciate some deeper in vivo analysis. Is the construct stable over time? What are expected side effects? Does the immune system target the construct? What happens when people overdose in their coffee consumption? Does the system then desensitize? All these points should at least be carefully discussed in the manuscript. The pharmacokinetics of the caffeine are further much different to GLP-1. This said, the half-life of GLP-1 is just 1-2 minutes while the caffeine is much longer in the circulation. How does this affect the acute and chronic effect of the GLP-1 and does this change over time and dependent of the lifestyle habits?

- the authors should discuss their opinion on the translational value of their system. Do the authors believe that patients can ever be safely implanted with such a caffeine-inducible construct? A short statement would be appreciated.

Reviewer #2 (Remarks to the Author):

In this manuscript, the authors developed a caffeine-inducible system named C-STAR by capitalizing on a caffeine specific single-domain antibody to directly sense caffeine at physiologically relevant concentrations in human cells and mouse models. This allows fine-tuning of both in vitro and in vivo transgene expression in response to pure caffeine as well beverages such as tea and coffee. The C-STAR system demonstrates great tenability and sensitivity and thus is a promising candidate for controlling Type-2 diabetes (T2D) and as an in vitro caffeine-quantifying device. Compared to formerly reported inducers (e.g. phloretin, benzoate, vanillic acid and blue light), caffeine is non-toxic, cheap and only present in specific beverages, making it an ideal small molecule inducer for therapeutic purposes. However, despite the impressive amount of work shown in this manuscript, there are doubts regarding the novelty of this study. I suggest the authors address the following points during revision.

Major points:

1. The overall design strategy and signaling pathway triggers are similar to reported works (Ref 16, 17 and 46). Also, caffeine consumption is not without 'potential side effects' (Page 2, line 50) and overdose is not unheard of. The authors need stronger justifications on the novelty of the strategy and the use of caffeine as an inducer.
2. Page 2 and line 130: Caffeine is a methylxanthine alkaloid and has several structural analogs such as theophylline, theobromine and paraxanthine which are present in common beverages as well. In the case of theophylline, past studies have shown that a single-domain VHH antibody against caffeine also showed a moderate binding affinity for theophylline (Refs 24-26). The authors have demonstrated the functionality of the C-STAR system with various concentrations of caffeine to support the sensitivity and reversibility of the system. However, there is a lack of data to show the specificity of the C-STAR system. It would be important to demonstrate that the C-STAR system is only activated by the presence of caffeine and not by the other compounds with similar structures. The authors should include data to exclude the possibilities of interference from caffeine analogs.
3. Page 4, line 132: The authors claimed that hMSC-hTERT cells yielded similar results (Fig. 2a) to those in HEK-293T cells (Fig. 1D). However, the values and trends in these two figures look drastically different. The authors should elaborate on the similarities that they are alluding to and justify their claims.
4. Page 5: The authors tested the designed C-STAR system in vitro in vascularized microcontainers with pure caffeine and in vivo by feeding T2D mice with regular oral doses of coffee. As caffeine is the inducer in the current work and regular oral doses are in fact difficult to define, there is a risk of hypoglycemia due to excessive release of shGLP1 if a high dose of caffeine is consumed. The authors should provide more data with the animal model to investigate the possible side effects of hypoglycemia upon high consumption of caffeinated beverages.
5. Page 5, Line 183: ShGLP1 protein was used in Ref 34 for the treatment of T2D, while the current work used shGLP1 coupled to mouse IgG as a fusion protein (shGLP1-mIgG). The authors should justify the purpose of using shGLP1-mIgG rather than shGLP1.
6. Page 7, line 255: The authors transfected HEK-293T cells with pSB100X (1900 ng)/ pDB326 (100 ng) to generate stable cell lines by the sleeping beauty transposon system for gene integration. Why was a 19:1 ratio of transposase to transposon cargo plasmid used? Can a lower ratio be just as effective? The high ratio may increase the possibility of random pSB100X

integration, which is unfavorable for therapeutic use. The authors should justify the need for using such a large amount of pSB100X for the transfection and investigate if a lower quantity of pSB100X plasmid is sufficient for integration.

7. Page 11, Line 406: The shGLP1 and insulin levels triggered by caffeine feeding were measured only after two days. The authors should also measure the shGLP1 and insulin levels a few hours after caffeine feeding to investigate the short-term effects.

Minor points:

1. Page 6, Line 222: To avoid overstating, I suggest rephrasing 'potential winner' to 'potential candidate'.

2. Page 7, line 255: The authors claimed that the stable designer cell lines were generated by co-transfection of pSB100X and pDB326 plasmid. However, the pDB326 plasmid did not contain a STAT3 responsive module, based on the information in the supplementary materials. Please clarify. The authors should also include more detailed information on the protocols used for transfecting the other plasmids (e.g. pLS13) to generate the stable cell lines.

3. Page 11, captions for Figure 3b-c, and captions for Supplementary Figure 3 and 6: The captions are highly repetitive. Please consider simplifying for improved readability.

4. Figure 3: Please convert the unit g/L to μM for consistency with the other figures to facilitate comparison.

5. Figure 3b and c: The x-axis labels are too cluttered and tedious to read. Please change the presentation to improve clarity.

Reviewers' comments

Reviewer #1

In this manuscript, Bojar and colleagues designed and evaluated a series of synthetic factors that upon transfection into the host cell induce a transient activation of targeting signaling events upon induction by caffeine. Using a series of elegant in vitro and in vivo approaches, the authors convincingly show caffeine-induction of targeted signaling events, suggesting that such a caffeine-induced approach has a broad pharmaceutical value for the treatment of various metabolic conditions.

The manuscript is well written, easy to follow and deals with a novel, innovative and important topic, the design and validation of specially-tailored molecules/constructs aimed to optimize the therapeutic value of certain drugs. The manuscript is considered as of importance for its field.

Main criticism

1. The authors state to have analyzed the caffeine-inducible GLP-1 construct in a mouse model for type 2 diabetes. The authors describe to have generated this T2DM model by repeated administration of streptocotozin (STZ). Notably, STZ is not inducing type 2 diabetes, it rather drives apoptosis of the beta cells and thus the development of hypoinsulinemia and type 1 diabetes. Type 2 diabetes is typically characterized by hyperinsulinemia and insulin resistance but the authors here see, as expected by STZ treatment, hypoinsulinemia and no insulin resistance. So this model can't be referred to as type 2 diabetic. Typically db/db mice or NZO mice are considered the best available mouse models for type 2 diabetes but not STZ treated mice.

We thank the reviewer for this important comment. We have followed this advice and have replaced the data of the STZ-induced diabetes model (old Fig. 4) with new data resulting from additional experiments on both DIO mice (new Fig. 4) and db/db mice (new Fig. 5), which are two well-established type-2 diabetic mouse models. As expected, both mouse models develop hyperinsulinemia (new Fig. 4C and new Fig. 5B), which could be corrected by C-STAR-controlled GLP-1-based treatment (new Fig. 4B and new Fig. 5A), leading to enhanced insulin production (new Fig. 4C and new Fig. 5B) and improved glucose tolerance (new Fig. 4D and new Fig. 5C).

2. Apart from the misclassification of the STZ mice as type 2 diabetic, these mice develop a somehow unexpected phenotype that warrants clarification. In detail, according to Figure 4f, fasting levels of blood glucose are roughly 145mg/dl (around 8 mM/L), which is still in the physiological range of non-STZ treated mice and is way lower as what is expected from a mouse that has repeatedly received an STZ treatment and which is hypoinsulinemic (as demonstrated in Figure 4d). Also, since GLP-1 decreases blood glucose via stimulation of insulin secretion, the insulinotropic action of GLP-1 is typically blunted in hypoinsulinemic type 1 diabetic STZ treated mice. To consolidate their findings, it is suggested that the authors repeat the in vivo study in an established mouse model for type 2 diabetes, like the db/db or the NZO mouse. Analysis of the construct in diet-induced obese (DIO) mice would further be of appreciable merit.

Please refer to our previous response. We have deleted the STZ data, and replaced it with new data sets obtained with both DIO mice (new Fig. 4) and db/db mice (new Fig. 5). We confirmed the appearance of hyperinsulinemia (new Fig. 4C and new Fig. 5B) and

characteristic hyperglycemia of T2D (**new Fig. 4A** and **new Fig. 5C**). Caffeine-triggered C-STAR-controlled insulinotropic action of GLP-1 was demonstrated in both mouse models of experimental Type-2 diabetes (**new Figs. 4B-C** and **new Figs. 5A-B**).

3. Does coffee-induced GLP-1 secretion decrease body weight in obese DIO mice? Especially the *in vivo* part can be substantially improved and would help to underline the pharmacological value of the system. Given the cardiovascular effect of caffeine, a control group of WT animals getting caffeine (but in which the caffeine can't activate the GLP-1 construct) would be important to add in these studies.

We have added new experiments assessing the impact of caffeine on the heart rate of wild-type, db/db and DIO mice (**new Fig. 4E** and **new Fig. 5D**), as well as profiling the body weight of C-STAR-treated DIO-mice treated with caffeine (**new Fig. 4F**). Our results show that caffeine-induced GLP-1 secretion significantly reduced the body weight of C-STAR-treated DIO mice (**new Fig. 4F**). Additionally, wild-type mice receiving caffeine in the absence of functional C-STAR implants showed no caffeine-dependent effects related to heart rate (**new Fig. 4E** and **new Fig. 5D**), body weight (**new Fig. 4F**), blood glucose (**new Fig. 4A**) or blood insulin (**new Fig. 4C**) over an extended experimental period of two weeks.

Minor criticism

1. Since the construct needs to be implanted into the host organism, I would appreciate some deeper *in vivo* analysis. Is the construct stable over time? What are expected side effects? Does the immune system target the construct? What happens when people overdose in their coffee consumption? Does the system then desensitize? All these points should at least be carefully discussed in the manuscript.

We have extended our experimental analysis of C-STAR-driven therapeutic effects on blood glucose (**new Fig. 4A**) and blood insulin (**new Fig. 4C**) to two weeks, a widely accepted standard for long-term analysis and validation of experimental disease models (Xie et al., 2016, *Science*, 354:1296; Wang et al., *Nat Biomed Eng*, 2:114; Tastanova et al., 2018, *Sci Transl Med*, in press). C-STAR-treated mice regularly consuming coffee showed significantly reduced (hyper)glycemia compared to mock-treated animals over the entire experimental period (**new Fig. 4A**), confirming that the system is stable over time and does not desensitize. Importantly, increased doses of coffee did not cause hypoglycemia (**new Supplemental Fig. S10**), confirming the established finding that GLP-1 is naturally inactive during normoglycemia (<10 mM; Meier et al., 2003, *J Clin Endocrinol Metab*, 88:2719; Doyle and Egan, 2007, *Pharmacol Ther*, 113:546). Additionally, control experiments profiling the cardiovascular impact of caffeine (**new Fig. 4E** and **new Fig. 5D**) and the immunogenicity of implanted microcapsules (**new Fig. S6**) did not show any side effects. We have updated the results and discussion sections of the revised manuscript accordingly.

2. The pharmacokinetics of the caffeine are further much different to GLP-1. This said, the half-life of GLP-1 is just 1-2 minutes while the coffee is much longer in the circulation. How does this affect the acute and chronic effect of the GLP-1 and does this change over time and dependent of the lifestyle habits?

We used the shGLP-1 variant, which contains a fused murine IgG domain that extends the half-life of natural GLP-1 (Ye et al., 2011, *Science*, 332:1565-8). We have also done a new pharmacokinetics experiment showing that the residence time of shGLP-1 in the circulation of mice is over 24 h following caffeine stimulation, whereas caffeine is more rapidly degraded

(**new Fig. S9**). These relative pharmacokinetics make caffeine an ideal trigger compound for cell-based (diabetes) therapy, enabling a transient stimulus (caffeine intake) to rapidly initiate a sustained therapeutic response (shGLP-1 production and action).

This pharmacokinetic profile (**new Fig. S9**) is fully consistent with the new data sets (**new Fig. 4A-E** and **new Fig. 5D** and **new Fig. S10**) showing that neither caffeine alone nor caffeine in combination with shGLP-1 negatively impacts on glucose or insulin homeostasis, or on the cardiovascular health of wild-type and diabetic mice in the longer term, further supporting the idea that C-STAR may indeed be compatible with a patient-centered and lifestyle-compatible therapy for diabetes.

3. The authors should discuss their opinion on the translational value of their system. Do the authors believe that patients can ever be safely implanted with such a caffeine-inducible construct? A short statement would be appreciated.

We have performed additional *in vivo* experiments confirming the immunoprotective function of our alginate-based microcapsules (**new Fig. S6**) and added a short paragraph on the clinical applicability of alginate-encapsulated human cells (ClinicalTrials.gov NCT01379729; Jacobs-Tulleneers-Thevissen et al., 2013, *Diabetologia* 56:1605) to the revised discussion.

Reviewer #2

In this manuscript, the authors developed a caffeine-inducible system named C-STAR by capitalizing on a caffeine-specific single-domain antibody to directly sense caffeine at physiologically relevant concentrations in human cells and mouse models. This allows fine-tuning of both *in vitro* and *in vivo* transgene expression in response to pure caffeine as well as beverages such as tea and coffee. The C-STAR system demonstrates great tenability and sensitivity and thus is a promising candidate for controlling Type-2 diabetes (T2D) and as an *in vitro* caffeine-quantifying device. Compared to formerly reported inducers (e.g. phloretin, benzoate, vanillic acid and blue light), caffeine is non-toxic, cheap and only present in specific beverages, making it an ideal small molecule inducer for therapeutic purposes. However, despite the impressive amount of work shown in this manuscript, there are doubts regarding the novelty of this study. I suggest the authors address the following points during revision.

1. The overall design strategy and signaling pathway triggers are similar to reported works (Ref 16, 17 and 46). Also, caffeine consumption is not without 'potential side effects' (Page 2, line 50) and overdose is not unheard of. The authors need stronger justifications on the novelty of the strategy and the use of caffeine as an inducer.

The C-STAR design is fundamentally different from previous work (refs. 16, 17, 46), in that the previous systems (refs. 16, 17, 46) use naturally evolved genetic components, such as bacterial transcription factors or human GPCRs as sensing modules. In contrast, the C-STAR system uses a completely novel, fully synthetic, non-natural gene switch, which is activated by caffeine-mediated protein dimerization and induces synthetic promoters in mammalian cells. A key advantage of our strategy is that it enables the design of synthetic gene switches tailored to sense trigger compounds (e.g., caffeine) for which no natural receptor exists. We have clarified this point in the revised manuscript (results section) and modified the schematic of **Fig. 1** accordingly (**new Fig. 1A**).

The issue of caffeine-induced side effects was also raised by reviewer 1. In response, we have added control experiments showing that caffeine does not cause significant or unexpected side effects on the cardiovascular system in the longer term (**new Fig. 4E** and **new Fig. 5D**). Similarly, we show that coffee consumption at higher doses did not cause hypoglycemia (**new Supplemental Fig. S10B**; please refer to our response to point 4 below for details).

2. Page 2 and line 130: Caffeine is a methylxanthine alkaloid and has several structural analogs such as theophylline, theobromine and paraxanthine which are present in common beverages as well. In the case of theophylline, past studies have shown that a single-domain VHH antibody against caffeine also showed a moderate binding affinity for theophylline (Refs 24-26). The authors have demonstrated the functionality of the C-STAR system with various concentrations of caffeine to support the sensitivity and reversibility of the system. However, there is a lack of data to show the specificity of the C-STAR system. It would be important to demonstrate that the C-STAR system is only activated by the presence of caffeine and not by the other compounds with similar structures. The authors should include data to exclude the possibilities of interference from caffeine analogs.

We agree and have done additional control experiments that confirm the caffeine specificity of the C-STAR system. Among theophylline, theobromine and paraxanthine, only theophylline showed marginal activation of C-STAR at the maximal examined concentration of 1 μ M; however, such a high level is very unlikely to be reached as a result of food intake (Hicks et al., 1996, *Food Res Int*, 29:325; Hackett et al., 2008, *J Anal Toxicol*, 32:695) (**new Fig. S4**).

3. Page 4, line 132: The authors claimed that hMSC-hTERT cells yielded similar results (Fig. 2a) to those in HEK-293T cells (Fig. 1D). However, the values and trends in these two figures look drastically different. The authors should elaborate on the similarities that they are alluding to and justify their claims.

We apologize for the misstatement. Indeed, transiently transfected HEK-293T cells (Fig. 1D) show higher caffeine sensitivity than hMSC-hTERT cells (Fig. 2A).

4. Page 5: The authors tested the designed C-STAR system in vitro in vascularized microcontainers with pure caffeine and in vivo by feeding T2D mice with regular oral doses of coffee. As caffeine is the inducer in the current work and regular oral doses are in fact difficult to define, there is a risk of hypoglycemia due to excessive release of shGLP1 if a high dose of caffeine is consumed. The authors should provide more data with the animal model to investigate the possible side effects of hypoglycemia upon high consumption of caffeinated beverages.

We conducted an additional experiment, which confirmed that mice consuming coffee at high doses did not show hypoglycemic side effects due to increased shGLP-1 release (**new Fig. S10**). Although blood active shGLP-1 increased with higher caffeine doses (**new Fig. S10A**), hypoglycemia was never observed (**new Fig. S10B**). This result is in agreement with previous reports demonstrating that GLP-1 remains naturally inactive during normoglycemia (<10 mM; Meier et al., 2003, *J Clin Endocrinol Metab*, 88:2719; Doyle and Egan, 2007, *Pharmacol Ther*, 113:546). (Please see also our response to minor point 1 of reviewer 1 above).

5. Page 5, Line 183: ShGLP1 protein was used in Ref 34 for the treatment of T2D, while the current work used shGLP1 coupled to mouse IgG as a fusion protein (shGLP1-mIgG). The authors should justify the purpose of using shGLP1-mIgG rather than shGLP1.

We apologize for the confusion. shGLP-1 already contains the murine IgG domain fused to the natural GLP-1 peptide, which is the construct used in this work and in the work by Ye et al., 2011 (Science, 332:1565). We have removed the additional mIgG term from the plasmid descriptions.

6. Page 7, line 255: The authors transfected HEK-293T cells with pSB100X (1900 ng)/ pDB326 (100 ng) to generate stable cell lines by the sleeping beauty transposon system for gene integration. Why was a 19:1 ratio of transposase to transposon cargo plasmid used? Can a lower ratio be just as effective? The high ratio may increase the possibility of random pSB100X integration, which is unfavorable for therapeutic use. The authors should justify the need for using such a large amount of pSB100X for the transfection and investigate if a lower quantity of pSB100X plasmid is sufficient for integration.

We are sorry that the labeling of the transfected plasmids amounts was reversed by mistake. In fact, the transfected amounts were pSB100X (100 ng) / pDB326 (1900 ng), a ratio that was suggested to provide an optimal balance between integration efficiency and host cell metabolism (Kowarz et al., 2015, Biotechnol J, 10:647). We have corrected this in the revised manuscript.

7. Page 11, Line 406: The shGLP1 and insulin levels triggered by caffeine feeding were measured only after two days. The authors should also measure the shGLP1 and insulin levels a few hours after caffeine feeding to investigate the short-term effects.

We have added the requested pharmacokinetic analysis showing that shGLP-1 is already efficiently produced 6 h after caffeine feeding (new **Fig. S9B**). Insulinotropic action of shGLP-1 was further observed in db/db mice when performing glucose tolerance tests (new **Fig. 5C**) and profiling insulin levels 12 hours after caffeine feeding (new **Fig. 5A-B**). (Please see also our response to minor point 2 of reviewer 1 above).

Minor points

1. Page 6, Line 222: To avoid overstating, I suggest rephrasing ‘potential winner’ to ‘potential candidate’.

Done as requested.

2. Page 7, line 255: The authors claimed that the stable designer cell lines were generated by co-transfection of pSB100X and pDB326 plasmid. However, the pDB326 plasmid did not contain a STAT3 responsive module, based on the information in the supplementary materials. Please clarify. The authors should also include more detailed information on the protocols used for transfecting the other plasmids (e.g. pLS13) to generate the stable cell lines.

C-STAR_{DB1} is a mono-transgenic cell line stably expressing the caffeine receptor from pDB326. Therefore, experiments related to STAT3-dependent SEAP expression involved transient transfection of pLS13 into CSTAR_{DB1}. Only for the T2D animal experiments did we

use the double transgenic cell line $CSTAR_{DB6}$ stably expressing both the caffeine receptor and P_{STAT3} -shGLP-1. We have clarified this point in the revised manuscript.

3. Page 11, captions for Figure 3b-c, and captions for Supplementary Figure 3 and 6: The captions are highly repetitive. Please consider simplifying for improved readability.

We have followed this advice and have shortened the captions of Fig. S3 and S6.

4. Figure 3: Please convert the unit g/L to μ M for consistency with the other figures to facilitate comparison.

Done as requested.

5. Figure 3b and c: The x-axis labels are too cluttered and tedious to read. Please change the presentation to improve clarity.

Done as requested.

REVIEWERS' COMMENTS:

Reviewer #1 (Remarks to the Author):

in the revised version of the manuscript, the authors provide now more sophisticated analysis that overall considerably strengthen the manuscript. The only point I still suggest to change is that the body weight data shown in Figure 4F should be shown as longitudinal line blots instead of just a bar graph showing the absolute weight loss after the chronic treatment. This will help the reader to understand how and when the weight loss kicks in and how the progression of weight loss is over time.

Reviewer #2 (Remarks to the Author):

The authors have diligently addressed all the questions raised earlier and I am satisfied with the quality of the work.

NCOMMS-17-16052A- Response to the remaining reviewers' comments

REVIEWERS' COMMENTS:

Reviewer #1 (Remarks to the Author):

in the revised version of the manuscript, the authors provide now more sophisticated analysis that overall considerably strengthen the manuscript. The only point I still suggest to change is that the body weight data shown in Figure 4F should be shown as longitudinal line blots instead of just a bar graph showing the absolute weight loss after the chronic treatment. This will help the reader to understand how and when the weight loss kicks in and how the progression of weight loss is over time.

We are glad for the favorable comments of the reviewer. Unfortunately, we only measured the weight of the mice at the beginning and the end of the experiment and are therefore unable to convert it into longitudinal line blots. We based our experimental design for this on previous studies (Rössger et al., 2013, Nat Commun, 4:2825; Ye et al., 2017, Nat Biomed Eng, 1:0005).

Reviewer #2 (Remarks to the Author):

The authors have diligently addressed all the questions raised earlier and I am satisfied with the quality of the work.

We appreciate the reviewer's favorable comment on our revised manuscript.